# Bone Growth and Development in Different Breeds of Piglets at the Early Age Is Associated with Mineral Deposition

**DOI:** 10.3390/ani15243536

**Published:** 2025-12-08

**Authors:** Md. Abul Kalam Azad, Chenjian Li, Yating Cheng, Bo Song, Sujuan Ding, Zhenlei Zhou, Qian Zhu, Xiangfeng Kong

**Affiliations:** 1Hunan Provincial Key Laboratory of Animal Nutritional Physiology and Metabolic Process, Institute of Subtropical Agriculture, Chinese Academy of Sciences, Changsha 410125, China; 2College of Veterinary Medicine, Nanjing Agriculture University, Nanjing 210095, China; 3College of Advanced Agricultural Sciences, University of Chinese Academy of Sciences, Beijing 100049, China

**Keywords:** bone characteristics, calcium, pig breed, phosphorus, weaning piglets

## Abstract

Skeletal development plays a vital role in the growth and productivity of farm animals. Healthy bones are essential for supporting the body, facilitating muscle attachment and body movement, and protecting vital organs from injury. This study systematically evaluated bone growth and development in three pig breeds—Chinese domestic (Taoyuan black pigs and Xiangcun black pigs) and commercial Duroc pigs—at an early age. The findings revealed that the bone growth rate of commercial piglets is faster compared to that of Chinese native piglets. Moreover, weaning stress may adversely affect bone growth and development, with variations observed among pig breeds. The findings provide potential guidance for optimizing mineral supplementation strategies in Chinese native pigs to improve early-stage bone growth and development.

## 1. Introduction

Weaning is the most critical stages in the life cycle of pigs, often leading to intestinal and immune system dysfunction [1,2]. Weaning transition often leads to reduced feed intake, intestinal inflammation, and impaired digestive and absorptive functions of the small intestine [3,4]. Regardless of the weaning age, piglets undergo weight loss and require approximately four days after weaning to recover to the original level [5]. Therefore, revealing the changes in the morphological structure and physiological functions of major organs in piglets before and after weaning is of great practical significance for alleviating the negative effects of weaning stress.

Skeletal development in piglets is highly dynamic and provide the structural foundation for their overall growth [6]. Bone growth is also one of the important indicators for evaluating the growth and development of piglets [7]. Bones are the reservoirs for minerals such as calcium (Ca) and phosphorus (P), releasing these elements into the bloodstream when the body’s demand increases, thereby maintaining the Ca-P homeostasis [8]. Bone morphology provides insight into the breed-specific growth trajectories and potential differences in biomechanical loading. Moreover, bone mineral density (BMD) and mineralization are critical indicators of skeletal health, reflecting the deposition of minerals such as Ca and P, which are essential for bone strength and resilience [9,10]. Therefore, clarifying the regulations of bone growth and mineral metabolism in early-aged piglets and their relationship can provide a theoretical basis for promoting pig growth, development, and overall health through nutritional strategies.

China has rich diversity of pig breed resources, with relatively close genetic distance despite the variations among breeds [11]. Native pig breeds have a high level of adaptability to local environments, unique genetic traits, and excellent meat quality. However, compared to commercial pig breeds such as Duroc, Chinese native pig breeds have slower growth rates and higher feed-to-meat ratios [12], resulting in a less competitive in breeding efficiency. The Taoyuan black (TB) pig, a local breed in Hunan province, is known for its high intramuscular fat content and resistance to roughage feeding [13]. The Xiangcun black (XB) pig, a crossbred pig variety, with Duroc pig as the paternal line and TB pig as the maternal line, combines traits from both breeds. Currently, the evaluation of pig breed quality mainly focuses on growth performance, meat quality, and other traits directly related to economic benefits, with relatively limited attention given to bone development. Therefore, this study aims to investigate the differences in bone growth and development between Chinese native piglets and Duroc piglets at different early age stages, with a focus on bone length, weight, and index, as well as BMD and mineralization. By comparing these parameters, this study was mainly aimed at elucidating breed-specific patterns of skeletal development and providing a foundation for further research into the genetic and environmental factors influencing bone health in piglets. The findings of this study may contribute to the development of targeted nutritional and management strategies to enhance skeletal development and overall productivity in pig farming.

## 2. Materials and Methods

### 2.1. Animal and Experimental Design

A total of 30 litters of newborn TB, XB, and Duroc piglets (ten litters/breed) with an average birth weight of the pig breed (TB, 1.50 ± 0.25 kg; XB, 1.39 ± 0.19 kg; and Duroc, 1.82 ± 0.43 kg) from their respective sows (sows with 9–11 piglets/litter) were randomly selected for this study. TB and XB sows were purchased from Xiangcun High-Tech Agricultural Co., Ltd., Loudi, China, and Duroc sows were purchased from Tianxin Breeding Co., Ltd., Changsha, China. The experimental piglets were reared in the same barn, with different pig breeds separate. The ambient environment for suckling piglets was maintained at a thermostatically controlled temperature of 30 °C for the first week, then reduced by 2 °C the following week until weaning. All piglets had free access to maternal milk until weaning (21 days old) and to starter feed after weaning at 8:00, 12:00, 14:00, and 17:00 daily. The composition and nutrient levels of the starter diets for the three different pig breeds met the Chinese local swine nutrient requirements (GB NY/T 65 2004) [14] and the National Research Council (NRC, 2012) diet requirements [15] (Appendix A). The experimental house was maintained at a control temperature (23–26 °C) with forced air ventilation. The experimental piglets did not receive any vaccinations during the trial.

### 2.2. Sample Collection

At 1, 10, 21 (weaning), and 24 (three days of post-weaning) days old, a total of 30 piglets (half male and half female), 10 piglets from each breed (one piglet from each litter), close to the average body weight (BW) of the litter, were randomly selected for sampling at each age stage. Blood samples (10 mL) were collected from the anterior vena cava of each piglet and transferred into heparin anticoagulant tubes. The samples were then centrifuged at 3500× *g* for 10 min to obtain the plasma for biochemical parameter analysis. After anesthesia with intramuscular injection of Zoletil^®^50 (Beijing Lab. Anim. Tech. Dev. Co., Ltd., Beijing, China) and exsanguination, the piglets were dissected to collect the left hind limb femur, tibia, fourth rib, and fourth lumbar vertebrae. Surface muscle tissues were carefully removed using the surgical scissor from all collected bones and then stored at −20 °C to further determine bone-related parameters.

### 2.3. Determination of Bone Parameters

The frozen bone samples were naturally thawed and moisture was removed with absorbent paper. Afterward, bone samples were weighed, and bone lengths were measured using the Vernier caliper (HengLiang Tech. Co., Ltd., Zhengzhou, China). The bone indexes of those bones were calculated using the following formula:Bone index (g/kg) = bone weight (g)/BW (kg)

The midpoint diameter of the femur, tibia, and rib was measured using a Vernier caliper (HengLiang Tech. Co., Ltd., Zhengzhou, China). Afterward, a quasi-static three-point bending test was performed using a universal testing machine with a measuring range of 1 N to 5000 N (Zwick Roell, Z5.0; Ulm, Germany) under the following parameters: a support span equal to 40% of the average bone length (rounded to the nearest millimeter) and a loading rate of 10 mm/min until fracture. The peak of maximum force was recorded. The NexygenPlus (v4.18) materials testing software (Ametek, Largo, FL, USA) was used to calculate the test data.

Dual-energy X-ray absorptiometry (DXA) was used to assess the BMD and bone mineral content (BMC) of those bones, as previously described, according to the methods of Li et al. [16].

### 2.4. Determination of Ash, Calcium, and Phosphorus Content in Bone

The femur, tibia, and rib (the lumbar vertebrae were discarded due to difficulty in cutting) were taken after drying at 70 °C for 12 h to assess the ash content. The detailed methods were adopted from Shao et al. [17]. The ash content is expressed as the percentage of the dry weight of the defatted bone.

The obtained ash samples were dissolved in 10 mL of a 6 mol/L HNO_3_ solution and then transferred in 100 mL volumetric flasks to measure the Ca and P content of bones, as previously described by Wang et al. [18].

### 2.5. Determination of Plasma Calcium, Phosphorus, and Alkaline Phosphatase Levels

The levels of Ca, P, and alkaline phosphatase (ALP) in plasma were measured with an automated biochemical analyzer (Roche, Basel, Switzerland) according to the manufacturer’s guidelines.

### 2.6. Statistical Analysis

The results are presented as means with SE values. Data normality was assessed using the Shapiro–Wilk test, and homogeneity of variance was evaluated using Levene’s test. The individual piglets were considered the experimental unit for all analyses. Statistical analyses were performed by a two-way ANOVA for pig breed and day of age using R software (version 4.2.1, R Core Team, 2022). The differences between the means of the experimental groups were assessed using the one-way ANOVA and Tukey’s post-hoc test for comparison. The differences among groups were considered significant when the *p*-values were <0.05. The correlations between Ca and P levels in bones and BMC and BMD were analyzed using Pearson correlation test with the R program.

## 3. Results

### 3.1. Changes in Bone Parameters of Piglets

As shown in Table 1, the lengths of the femur and tibia in TB and XB piglets at 1, 10, 21, and 24 days old were smaller (*p* < 0.05) than that of Duroc piglets. The rib length of TB and XB piglets was smaller (*p* < 0.05) at 10 and 24 days old, as well as the length of lumbar vertebrae at 10, 21, and 24 days old, compared to Duroc piglets. Compared with TB piglets, the bone lengths of the four bones in XB piglets were longer at 21 days old, whereas the rib in XB piglets was longer at 24 days old (*p* < 0.05). In the comparisons among the same breed of piglets at different ages, the bone lengths of the four bones in each pig breed were increased (*p* < 0.05) with the age. At 21 and 24 days old, the tibia length of Duroc and XB piglets and the femur of XB and TB piglets were longer, whereas the rib and lumbar vertebrae of XB piglets were smaller than those at 1 and 10 days old (*p* < 0.05).

As presented in Table 2, the femur and tibia weights of XB and TB piglets at four different ages were lighter, and the rib weight at 10 and 24 days old and lumbar vertebrae at 10, 21, and 24 days old were also lighter compared to Duroc piglets (*p* < 0.05). Moreover, the weights of rib and lumbar vertebrae in XB piglets at 1 day old and the bone weights of the four bones in XB piglets at 21 and 24 days old were lighter (*p* < 0.05) than those of the TB piglets. Among the same breed of piglets at different ages, the bone weights of the four bones showed an increasing trend. At 21 and 24 days old, the bone weights of the four bones in Duroc piglets were heavier, and the femur, tibia, and lumbar vertebrae weights of XB and TB piglets were also heavier than those at 1 and 10 days old (*p* < 0.05). Additionally, the rib weight of XB piglets was heavier (*p* < 0.05) at 10 days old than at 1 day old. The rib weight of TB piglets was heavier at 10 and 21 days old, but it was lighter at 21 and 24 days old (*p* < 0.05).

As indicated in Table 3, the femur and tibia indexes of TB and XB piglets at 1, 10, and 24 days old, the rib indexes at all four ages, and the lumbar vertebrae index at 24 days old were lower (*p* < 0.05) compared with those of Duroc piglets. The lumbar vertebrae index of XB piglets at 1 day old and of TB piglets at 21 days old were lower (*p* < 0.05) compared with those of Duroc piglets. The lumbar vertebrae index of TB piglets was lower (*p* < 0.05) at 10 days old compared with Duroc and XB piglets. Among different ages within the same breed of piglets, the bone indexes of the four bones in Duroc piglets decreased (*p* < 0.05) at 1 and 10 days old. The femur index of XB piglets at 24 days old was higher compared to that at 1 day old, and the lumbar vertebrae index of XB piglets increased at 1 and 10 days old (*p* < 0.05). There were no significant differences in the bone indexes of the four bones in TB piglets among all ages (*p* > 0.05).

### 3.2. Changes in Bone Breaking Load, Bone Mineral Content, and Bone Mineral Density of Piglets

As listed in Table 4**,** the bone breaking load (BBL) of the femur at 10 and 21 days old and tibia and rib at 10 days old were lower (*p* < 0.05) in XB and TB piglets compared to the Duroc piglets. The BBLs of the femur at 24 days old and the tibia and rib at 21 days old were lower (*p* < 0.05) in XB piglets compared to Duroc piglets. Compared to TB piglets, the BBL of the femur was lower (*p* < 0.05) at 1 day old in Duroc and XB piglets. Moreover, the BBLs of the tibia at 24 days old were higher (*p* < 0.05) in Duroc and TB piglets than in XB piglets. Among different ages in the same pig breed, the BBLs of tibia and rib at 1 and 10 days old, and the femur and tibia at 10 and 21 days old, were higher (*p* < 0.05) in Duroc piglets. In XB piglets, the BBLs of the tibia at 1 and 10 days old, and the femur and tibia at 10 and 21 days old, were higher (*p* < 0.05) compared to that at 1 day old. The BBL of the rib in XB piglets was higher (*p* < 0.05) at 24 days old compared to that at 1 day old. Moreover, the BBLs of the femur, tibia, and rib in TB piglets were higher (*p* < 0.05) at 10 and 21 days old compared with at 1 day old.

As shown in Table 5, the mineral content in the femur and tibia of XB piglets at 1 day old and XB and TB piglets at 10, 21, and 24 days old was lower (*p* < 0.05) than that of the Duroc piglets. The mineral content in the rib and lumbar vertebrae of XB and TB piglets at 10 and 24 days old was lower (*p* < 0.05) than that of the Duroc piglets. The mineral content in the rib and lumbar vertebrae of TB piglets at 1 day old was higher (*p* < 0.05), as well as in the femur and tibia of TB piglets at 21 and 24 days old, than that of the XB piglets. Moreover, the mineral content in the rib and lumbar vertebrae of Duroc and TB piglets at 21 days old was higher (*p* < 0.05), as well as in TB piglets at 24 days old, than in the XB piglets. Among different ages in the same pig breed, the mineral content in four bones of Duroc and XB piglets was higher (*p* < 0.05) at 10, 21, and 24 days old than at 1 day old. The mineral content in the femur, tibia, and lumbar vertebrae of TB piglets was higher (*p* < 0.05) at 21 days old than at the other three ages. However, the mineral content in the rib of TB piglets was higher (*p* < 0.05) at 21 and 24 days old compared with at 1 and 10 days old.

As shown in Table 6, the BMD in the rib at 1 day old of XB and TB piglets was higher than that of Duroc piglets, whereas the BMD in the femur and tibia was lower at 21 and 24 days old compared to Duroc piglets (*p* < 0.05). Compared to TB piglets, the BMD in the ribs and lumbar vertebrae of Duroc and XB piglets was lower (*p* < 0.05) at 21 days old. Moreover, the BMD in the lumbar vertebrae of Duroc and TB piglets was higher (*p* < 0.05) than that of XB piglets. When comparing different ages within the same pig breed, the BMD in the femur, tibia, and ribs of Duroc piglets was higher (*p* < 0.05) at 10, 21, and 24 days old compared to that at 1 day old, whereas the BMD in the lumbar vertebrae of Duroc piglets was higher (*p* < 0.05) at 24 days old compared to 1 and 10 days old. In XB piglets, the BMD in the femur was higher at 10 and 21 days old, as well as in the tibia at 10, 21, and 24 days old compared to that at 1 day old (*p* < 0.05). For TB piglets, the BMD in all four bones was higher (*p* < 0.05) at 21 days old compared to that at 1 and 10 days old.

### 3.3. Changes in Ash, Calcium, and Phosphorous Levels

As shown in Figure 1, the ash content in the tibia of XB and TB piglets at 1 day old was lower (*p* < 0.05), as well as in the femur of XB and TB piglets at 10 days old, than in Duroc piglets. The ash content in the femur and tibia of XB and TB piglets at 21 days old was higher (*p* < 0.05) compared to that of Duroc piglets. At 24 days old, the ash content in the femur and tibia of TB piglets was lower (*p* < 0.05), as well as in the ribs of XB piglets, than that of Duroc piglets. Additionally, the ash content in the tibia of XB piglets at 1 day old, and in the ribs at 10 and 24 days old, was lower (*p* < 0.05) compared to that of TB piglets.

As shown in Figure 2A,B, at 1 day old, the Ca content in the femur and ribs of XB piglets was higher (*p* < 0.05) than that in Duroc piglets. The Ca and P levels in the tibia of XB and TB piglets at 10 days old were lower (*p* < 0.05) compared to those in Duroc piglets. At 21 days old, the Ca and P levels in the femur and tibia of XB and TB piglets were lower (*p* < 0.05) than those in Duroc piglets. Compared to Duroc piglets, the Ca and P levels were higher (*p* < 0.05) in the tibia of TB piglets at 24 days old. Additionally, compared to TB piglets, the Ca level in the tibia at 1 day old, as well as in the femur at 24 days old were higher (*p* < 0.05) in XB piglets.

### 3.4. Changes in Plasma Calcium, Phosphorus, and Alkaline Phosphatase Levels

According to Table 7, plasma Ca and P levels in XB and TB piglets were lower (*p* < 0.05) at 1 day old compared to those in Duroc piglets. Compared to TB piglets, plasma Ca and P levels in Duroc and XB piglets were lower (*p* < 0.05) at 21 days old. When comparing different ages within the same pig breed, the plasma Ca level in Duroc piglets decreased, while the plasma P level increased from 1 to 10 days old (*p* < 0.05). Additionally, the plasma P level in Duroc piglets decreased (*p* < 0.05) from 21 to 24 days old. In XB piglets, the plasma P level increased from 1 to 10 days old but decreased continuously from 10 to 24 days old (*p* < 0.05). For TB piglets, the plasma Ca level at 21 days old was higher than that at 1 day old, while the plasma P level was lower at 1 and 24 days old compared to at 10 and 24 days old (*p* < 0.05).

As presented in Table 8, the plasma ALP level was higher (*p* < 0.05) in XB piglets at 10 days old than that in TB piglets. When comparing different ages within the same pig breed, the plasma ALP level of all pig breeds showed a decreasing trend with age (*p* < 0.05). Specifically, in Duroc piglets, the plasma ALP level declined from 1 to 10 days old (*p* < 0.05), whereas that in Duroc and TB piglets was even more declined (*p* < 0.05) at 24 days old compared with the other three age groups.

### 3.5. Correlations Between Calcium and Phosphorus Levels in Bones with BMD and BMC

The correlations between Ca and P levels in bones with BMD and BMC are shown in Figure 3. In Duroc piglets (Figure 3A), P level in the tibia at 10 and 24 days old was negatively correlated (*p* < 0.05) with BMC of all bones, whereas the Ca level of the tibia at 24 days old was negatively correlated (*p* < 0.05) with the BMC of the ribs. Moreover, Ca and P levels of the ribs were positively correlated (*p* < 0.05) with the BMD of the ribs of Duroc piglets at 24 days old. However, no significant correlations (*p* > 0.05) were observed between Ca and P levels in bones and the BMD and BMC of Duroc piglets at 1 and 21 days old.

In XB piglets (Figure 3B), the P level in the femur was positively correlated (*p* < 0.05) with BMC of the ribs, whereas the P level in the tibia was positively correlated (*p* < 0.05) with BMD of tibia at 1 day old. Additionally, the Ca level in the ribs was positively correlated (*p* < 0.05) with the BMD and BMC of lumbar vertebrae at 21 days old. However, no significant correlations (*p* > 0.05) were observed between Ca and P levels in bones and the BMD and BMC of XB piglets at 10 and 24 days old.

In TB piglets (Figure 3C), a positive correlation (*p* < 0.05) was observed between the P level of the femur and the BMC of the ribs at 1 day old. At 10 days old, Ca and P levels of the femur were positively correlated (*p* < 0.05) with the BMC and BMD of the femur and tibia, as well as the BMD of lumbar vertebrae. Moreover, the P level of rib was positively correlated (*p* < 0.05) with the BMC of lumbar vertebrae at 24 days old. However, no significant correlations (*p* > 0.05) were observed between Ca and P levels in bones and BMD and BMC of TB piglets at 21 days old.

## 4. Discussion

Weaning is a crucial period in modern intensive farming, characterized by physiological and environmental changes for piglets. In this period, piglets face several challenges, including early organ development, diet changes (separates from mother’s milk to solid feed), and various physiological (e.g., transportation and mixing with other littermates) and environmental stressors (e.g., changes in ambient temperature). These factors often lead to decreased food intake, diarrhea, BW loss, and even death. In addition, intrauterine growth retardation and high culling rates during the weaning period have become prominent issues in commercial pig farming, affecting the economic benefits of the industry [19]. Although bone development in pigs occurs over a relatively long cycle and may not exhibit immediate changes, as piglets grow rapidly, the morphological changes in their bones within a few days still have a certain significance [7,20]. To better understand these changes, this study compared bone growth and development in different pig breeds at four key ages before and after weaning. The findings revealed that the bone growth rate of commercial piglets is relatively faster than that of the Chinese native pig breeds. The bone mechanical properties of TB piglets at 21 and 24 days old are relatively better than those of XB piglets. Moreover, this study also highlighted that weaning has a certain negative impact on the bone development of piglets.

In this current study, the bone growth rate of TB and XB piglets was significantly lower than that of Duroc piglets, as evidenced by their significantly shorter bone length and lighter bone weight at the same age. This result might be due to their BW at birth. Our previous studies have shown that the birth weight of XB piglets is lower than that of Duroc piglets [21]. When comparing the bone length data of different pig breeds at 1 day old, it was found that the bone lengths of the four major bones of the TB and XB piglets were smaller than those of Duroc piglets. Our previous study also indicated that the BW of TB and XB piglets is comparatively lower than that of Duroc piglets [21], which may explain why the bone length of these native pig breeds is smaller than that of Duroc pigs. In addition, we previously found that Duroc pigs have higher bone growth during the growing–finishing period compared to Chinese native pigs [16]. Furthermore, compared with TB piglets, the lengths of the femur, tibia, and lumbar vertebrae in XB piglets were significantly smaller at 21 and 24 days old. Additionally, the bone weight of XB piglets was significantly smaller at 21 and 24 days old, except for that of the ribs. These results indicate that the bone growth rate of XB piglets slows down at 21 days old compared to 24 days old, potentially due to weaning stress, which may lead to a decrease in bone growth in XB piglets. Research evidence has indicated that weaning stress induces inflammatory processes in the small intestine and hinders nutrient and mineral absorption, leading to deficiencies in mineral metabolism [22]. Moreover, our previous findings indicated that TB piglets exhibit greater immune tolerance than XB piglets at weaning, which may be due to the Chinese native breed having better abilities to adapt to the local environment than the cross-breed [23]. The above-mentioned features may be the possible reasons for the decreased bone growth in XB piglets at weaning. The bone index, representing the proportional weight of different bones relative to BW, was significantly higher in Duroc piglets compared to TB and XB piglets. However, no significant differences in the bone index were observed between TB and XB piglets at different ages. These findings indicate that Duroc piglets exhibit superior skeletal growth compared to the Chinese native pig breeds, which may be due to factors influencing early bone development, including the duration of prenatal development and body size at birth [7]. Moreover, genetic background is another possible reason for the greater bone growth and faster development of Duroc pigs, as evidenced by their higher feed efficiency and mineral absorption capacity compared with Chinese native pigs at these age stages [21].

The DXA is a commonly used method for bone mineral analysis, particularly for osteoporosis diagnosis. The BMD measured by DXA serves as a reliable indicator for assessing the risk of fracture and evaluating bone structure mechanics [24]. The BBL reflects the biomechanical integrity of bones and provides a more direct mechanical measure compared to structural characteristics such as the trabeculae arrangement [25]. Besides its biomechanical role, the BBL is important for supporting BW, enabling movement and protecting internal organs [26]. In this current study, there were no significant differences in the BMD or BBL of Duroc piglets at 1 day old compared to that of Chinese native piglets. However, the BBL of the femur in Duroc piglets was even significantly lower than that of TB piglets. At 10 days old, Duroc piglets exhibited a higher BBL, but no significant differences in bone density were detected. This increase in the BBL may be due to bone mass accumulation during early development. Overall, the bone mechanical properties of XB piglets, both before and after weaning, were relatively poor. This was evidenced by their lower bone length and weight compared to Duroc piglets. Moreover, TB piglets exhibited BBL and density characteristics that were comparable to, or in some cases superior to, those of Duroc piglets, which may be due to their greater ability to adapt to the local environment and their greater immune tolerance during weaning. Furthermore, correlation analysis revealed that the P level in the tibia of Duroc piglets was negatively correlated with the BMC of all four bones at 10 days old, whereas Ca and P levels in the femur of TB piglets were positively correlated with the BMC and BMD of the femur and tibia at 10 days old, indicating better mineral metabolism strengthening the bones of this pig breed. The compensatory advantage in bone quality may partially offset their slower apparent growth rate.

The composition of Ca and P in skeletal tissue certainly accounts for the majority of Ca and P in the body. Thus, measuring bone ash content has been established as a standard procedure for determining Ca and P availability or deposition, which is also an indicator of bone mineralization [27]. In this current study, the ash content in the femur of Duroc piglets was higher at 10 days old, while that in the femur and tibia, as well as the mineralization levels, were higher in Chinese native pig breeds at 21 days old. Three days after weaning (24 days old), the differences in bone ash content among the three pig breeds diminished, and the trends in bone ash content between Duroc and Chinese native pig breeds at different ages were not consistent. Higher levels of bone mineralization are associated with a relative reduction in the organic components of the bone [28]. Compared with Duroc piglets, Chinese native pig breeds displayed faster deposition of hydroxyapatite crystals and relatively slower syntheses of collagen fibers at the end of the lactation period. These findings suggest distinct patterns of bone development among different pig breeds.

Regarding the plasma Ca and P levels, pig breeds with a higher plasma Ca level also exhibited a higher plasma P level. Compared with Chinese native pig breeds, the plasma Ca and P levels were comparatively higher in Duroc piglets at 1 day old, whereas those of TB piglets at 21 days old were significantly higher than those of XB and Duroc piglets. This difference can mainly be explained by the fact that the birth weight of Duroc pigs is higher than that of Chinese native pigs [29], which have a better nutrient absorption ability and mineralization capacity. On the other hand, Duroc and XB pigs are more susceptible to weaning stress than TB piglets [23], including changes in diet type and immune resistance during weaning, which in turn affects mineral deposition. The differences in ALP level were varied among the three pig breeds, and no significant differences were observed between these breeds. Plasma ALP, Ca, and P levels are regulated by several complex factors and are not suitable for assessing the bioavailability of Ca and P elements alone [30]. In this current study, the different trends of Ca and P levels in plasma and bone were inconsistent, and no direct relationship was found between Ca and P levels in plasma and in bone.

When comparing different ages, the morphological and structural parameters of bones before weaning (1 and 10 days old) showed an increasing trend with age, while there was a delay in bone growth after three days of weaning (24 days old). However, no significant differences were observed in growth parameters. Three days after weaning, the bone density of XB and TB piglets showed a downward trend, while the bone density of Duroc piglets showed a slight increase, but there were no significant differences. Overall, these findings suggest that bone growth and development of Duroc piglets differed from the Chinese native piglets.

## 5. Conclusions

The bone growth rate of Duroc piglets is faster than that of XB and TB piglets. The mechanical properties of bone were superior in TB piglets at 21 and 24 days old compared to those of XB piglets. There are differences in the bioavailability of Ca and P among different pig breeds before and after weaning, while weaning stress may have a certain negative impact on the bone development of piglets. These findings provide a basis for the research of Chinese native pig breeds, which should be accelerated through nutritional strategies to improve the growth and development of Chinese native piglets during their early life.

## Figures and Tables

**Figure 1 animals-15-03536-f001:**
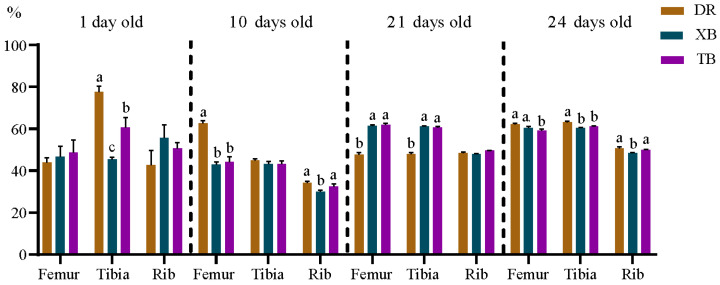
Differences in ash content in bones of Duroc, XB, and TB piglets before and after weaning (*n* = 6). Different lowercase letters indicate significant differences at *p* < 0.05. DR, Duroc; XB, Xiangcun black; TB, Taoyuan black.

**Figure 2 animals-15-03536-f002:**
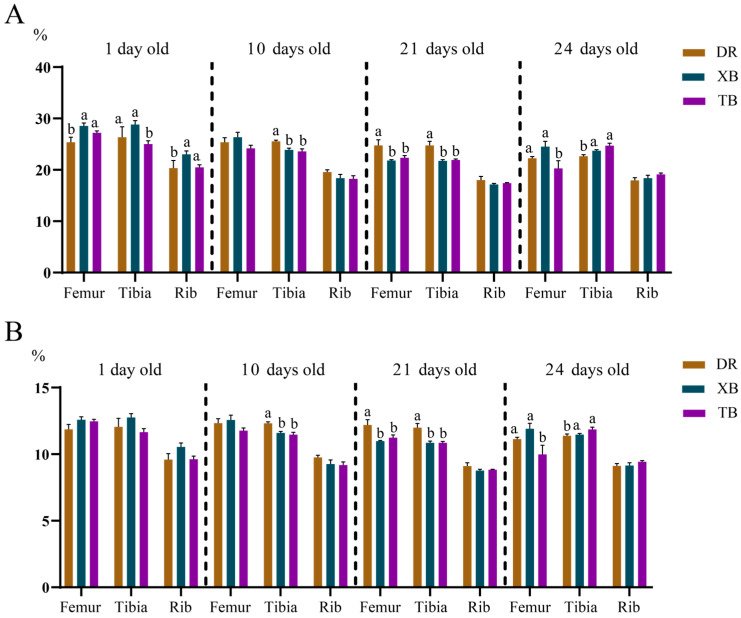
Differences in Ca (**A**) and P (**B**) content in the bones of Duroc, XB, and TB piglets before and after weaning (*n* = 6). Different lowercase letters indicate significant differences at *p* < 0.05. DR, Duroc; XB, Xiangcun black; TB, Taoyuan black.

**Figure 3 animals-15-03536-f003:**
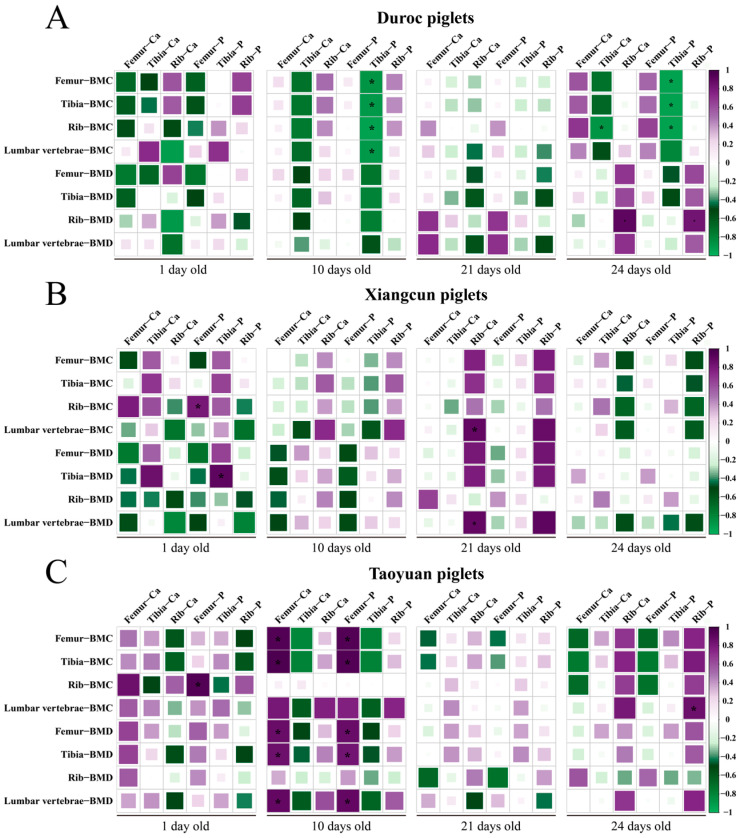
Pearson correlation analysis of Ca and P levels in bones with BMC and BMD before and after weaning. Correlation between Ca and P levels in bones and BMC and BMD of Duroc (**A**), Xiangcun (**B**), and Taoyuan (**C**) piglets. * *p* < 0.05.

**Table 1 animals-15-03536-t001:** Changes in bone length of piglets before and after weaning (cm).

Item	Duroc Piglet	XB Piglet	TB Piglet
Femur			
1 day old	51.78 ± 2.04 ^Ca^	47.69 ± 0.50 ^Db^	47.27 ± 1.17 ^Db^
10 days old	71.03 ± 1.73 ^Ba^	60.98 ± 0.67 ^Cb^	60.78 ± 0.75 ^Cb^
21 days old	81.04 ± 1.63 ^Aa^	66.13 ± 0.71 ^Bc^	74.68 ± 1.21 ^Ab^
24 days old	77.51 ± 1.51 ^Aa^	69.12 ± 0.91 ^Ab^	65.54 ± 1.86 ^Bb^
Tibia			
1 day old	56.27 ± 1.86 ^Da^	48.72 ± 0.40 ^Db^	51.57 ± 1.17 ^Cb^
10 days old	65.38 ± 1.27 ^Ca^	55.65 ± 0.84 ^Cb^	56.84 ± 0.41 ^Bb^
21 days old	76.05 ± 1.75 ^Ba^	62.90 ± 0.75 ^Bc^	69.84 ± 0.69 ^Ab^
24 days old	81.98 ± 1.63 ^Aa^	65.13 ± 0.87 ^Ab^	64.89 ± 1.45 ^Ab^
Rib			
1 day old	50.59 ± 1.72 ^C^	51.78 ± 1.05 ^C^	53.58 ± 0.93 ^D^
10 days old	65.27 ± 1.19 ^Ba^	56.16 ± 1.86 ^Bb^	57.01 ± 1.20 ^Cb^
21 days old	73.03 ± 1.70 ^Aa^	60.70 ± 1.39 ^Ab^	70.21 ± 1.30 ^Aa^
24 days old	71.43 ± 1.07 ^Aa^	62.35 ± 1.28 ^Ac^	65.86 ± 1.09 ^Bb^
Lumbar vertebrae			
1 day old	8.36 ± 0.36 ^C^	7.57 ± 0.26 ^C^	8.11 ± 0.30 ^D^
10 days old	11.89 ± 0.37 ^Ba^	9.67 ± 0.21 ^Bb^	10.31 ± 0.27 ^Cb^
21 days old	14.12 ± 0.61 ^Aa^	11.19 ± 0.26 ^Ac^	12.76 ± 0.23 ^Ab^
24 days old	13.33 ± 0.41 ^Aa^	11.96 ± 0.37 ^Ab^	11.42 ± 0.37 ^Bb^

Values are presented as means with SE (*n* = 10). Mean values with different lowercase superscript letters in the same row indicate significant differences (*p* < 0.05) among different pig breeds at the same age, while mean values with different uppercase superscript letters in the same column indicate significant differences (*p* < 0.05) among different ages of the same breed. XB, Xiangcun black; TB, Taoyuan black.

**Table 2 animals-15-03536-t002:** Changes in bone weight of piglets before and after weaning (g).

Item	Duroc Piglet	XB Piglet	TB Piglet
Femur			
1 day old	10.55 ± 1.14 ^Ca^	6.69 ± 0.23 ^Cb^	7.99 ± 0.46 ^Cb^
10 days old	23.32 ± 1.98 ^Ba^	12.73 ± 0.41 ^Bb^	13.17 ± 0.40 ^Bb^
21 days old	32.94 ± 2.45 ^Aa^	16.98 ± 0.86 ^Ac^	24.84 ± 0.76 ^Ab^
24 days old	35.49 ± 2.32 ^Aa^	16.81 ± 0.80 ^Ac^	22.71 ± 1.37 ^Ab^
Tibia			
1 day old	7.57 ± 0.94 ^Ca^	4.76 ± 0.19 ^Cb^	5.78 ± 0.36 ^Cb^
10 days old	15.85 ± 1.30 ^Ba^	8.93 ± 0.34 ^Bb^	9.36 ± 0.27 ^Bb^
21 days old	22.40 ± 1.56 ^Aa^	11.83 ± 0.59 ^Ac^	17.42 ± 0.55 ^Ab^
24 days old	24.55 ± 1.49 ^Aa^	11.39 ± 0.56 ^Ac^	15.67 ± 0.98 ^Ab^
Rib			
1 day old	1.19 ± 0.13 ^Ca^	0.68 ± 0.03 ^Bb^	1.09 ± 0.11 ^Ca^
10 days old	2.40 ± 0.19 ^Ba^	1.31 ± 0.12 ^Ab^	1.44 ± 0.08 ^Cb^
21 days old	3.57 ± 0.38 ^Aa^	1.59 ± 0.10 ^Ab^	2.96 ± 0.14 ^Aa^
24 days old	3.26 ± 0.30 ^Aa^	1.38 ± 0.12 ^Ac^	2.46 ± 0.18 ^Bb^
Lumbar vertebrae			
1 days old	2.12 ± 0.20 ^Ca^	1.31 ± 0.10 ^Cb^	1.79 ± 0.08 ^Ca^
10 days old	4.84 ± 0.40 ^Ba^	3.02 ± 0.14 ^Bb^	2.47 ± 0.20 ^Bb^
21 days old	7.18 ± 0.55 ^Aa^	3.81 ± 0.22 ^Ac^	5.06 ± 0.20 ^Ab^
24 days old	7.62 ± 0.48 ^Aa^	3.88 ± 0.19 ^Ac^	5.17 ± 0.30 ^Ab^

Values are presented as means with SE (*n* = 10). Mean values with different lowercase superscript letters in the same row indicate significant differences (*p* < 0.05) among different pig breeds at the same age, while mean values with different uppercase superscript letters in the same column indicate significant differences (*p* < 0.05) among different ages of the same breed. XB, Xiangcun black; TB, Taoyuan black.

**Table 3 animals-15-03536-t003:** Changes in bone index of piglets before and after weaning (g/kg).

Item	Duroc Piglet	XB Piglet	TB Piglet
Femur			
1 day old	7.57 ± 0.79 ^Aa^	4.57 ± 0.26 ^Bb^	5.19 ± 0.87 ^b^
10 days old	6.07 ± 0.08 ^Ba^	4.86 ± 0.13 ^ABb^	4.98 ± 0.12 ^b^
21 days old	5.55 ± 0.27 ^B^	5.05 ± 0.18 ^AB^	4.95 ± 0.18
24 days old	6.16 ± 0.14 ^Ba^	5.26 ± 0.17 ^Ab^	5.25 ± 0.19 ^b^
Tibia			
1 day old	5.42 ± 0.66 ^Aa^	3.25 ± 0.20 ^b^	3.76 ± 0.65 ^b^
10 days old	4.14 ± 0.07 ^Ba^	3.40 ± 0.10 ^b^	3.54 ± 0.08 ^b^
21 days old	3.78 ± 0.18 ^B^	3.52 ± 0.12	3.47 ± 0.12
24 days old	4.27 ± 0.09 ^Ba^	3.56 ± 0.10 ^b^	3.62 ± 0.13 ^b^
Rib			
1 day old	0.86 ± 0.09 ^Aa^	0.47 ± 0.02 ^b^	0.69 ± 0.12 ^ab^
10 days old	0.63 ± 0.03 ^Ba^	0.50 ± 0.04 ^b^	0.55 ± 0.03 ^ab^
21 days old	0.60 ± 0.05 ^Ba^	0.47 ± 0.03 ^b^	0.58 ± 0.02 ^a^
24 days old	0.56 ± 0.03 ^Ba^	0.43 ± 0.03 ^b^	0.57 ± 0.03 ^a^
Lumbar vertebrae			
1 day old	1.51 ± 0.12 ^Aa^	0.89 ± 0.07 ^Bb^	1.15 ± 0.18 ^ab^
10 days old	1.26 ± 0.03 ^Ba^	1.15 ± 0.05 ^Aa^	0.94 ± 0.07 ^b^
21 days old	1.21 ± 0.07 ^Ba^	1.13 ± 0.04 ^Aab^	1.00 ± 0.03 ^b^
24 days old	1.33 ± 0.05 ^ABa^	1.21 ± 0.02 ^Ab^	1.20 ± 0.05 ^b^

Values are presented as means with SE (*n* = 10). Mean values with different lowercase superscript letters in the same row indicate significant differences (*p* < 0.05) among different pig breeds at the same age, while mean values with different uppercase superscript letters in the same column indicate significant differences (*p* < 0.05) among different ages of the same breed. XB, Xiangcun black; TB, Taoyuan black.

**Table 4 animals-15-03536-t004:** Changes in bone breaking load of piglets before and after weaning (N).

Item *	Duroc Piglet	XB Piglet	TB Piglet
Femur			
1 day old	303.70 ± 22.67 ^Bb^	257.09 ± 19.57 ^Cb^	342.74 ± 36.59 ^Ba^
10 days old	389.22 ± 33.28 ^Ba^	299.26 ± 12.73 ^BCb^	294.47 ± 17.02 ^Bb^
21 days old	595.58 ± 50.52 ^Aa^	356.94 ± 29.47 ^Ab^	433.19 ± 21.76 ^Ab^
24 days old	560.56 ± 43.79 ^Aa^	342.87 ± 13.66 ^ABb^	423.51 ± 25.46 ^Aab^
Tibia			
1 day old	252.14 ± 23.53 ^C^	206.38 ± 14.85 ^C^	242.09 ± 25.51 ^B^
10 days old	382.11 ± 41.04 ^Ba^	274.49 ± 16.21 ^Bb^	259.09 ± 20.15 ^Bb^
21 days old	525.17 ± 54.79 ^Aa^	365.94 ± 27.91 ^Ab^	468.76 ± 23.21 ^Aab^
24 days old	542.48 ± 40.15 ^Aa^	328.66 ± 16.02 ^ABb^	489.17 ± 31.29 ^Aa^
Rib			
1 day old	17.45 ± 1.30 ^B^	14.04 ± 1.21 ^C^	13.99 ± 1.33 ^B^
10 days old	27.93 ± 2.23 ^Aa^	16.65 ± 1.27 ^BCb^	15.79 ± 1.08 ^Bb^
21 days old	33.90 ± 4.80 ^Aa^	21.77 ± 1.94 ^ABb^	25.18 ± 1.96 ^Aab^
24 days old	30.33 ± 3.09 ^A^	23.63 ± 2.12 ^A^	30.82 ± 3.26 ^A^

Values are presented as means with SE (*n* = 10). Mean values with different lowercase superscript letters in the same row indicate significant differences (*p* < 0.05) among different pig breeds at the same age, while mean values with different uppercase superscript letters in the same column indicate significant differences (*p* < 0.05) among different ages of the same breed. XB, Xiangcun black; TB, Taoyuan black. * Lumbar vertebrae samples were not collected due to difficulties in cutting.

**Table 5 animals-15-03536-t005:** Changes in bone mineral content of piglets before and after weaning (g).

Item	Duroc Piglet	XB Piglet	TB Piglet
Femur			
1 day old	1.48 ± 0.15 ^Ca^	1.11 ± 0.05 ^Cb^	1.31 ± 0.12 ^Dab^
10 days old	3.03 ± 0.29 ^Ba^	1.87 ± 0.10 ^Bb^	1.90 ± 0.10 ^Cb^
21 days old	4.95 ± 0.51 ^Aa^	2.49 ± 0.21 ^Ac^	3.79 ± 0.17 ^Ab^
24 days old	5.16 ± 0.36 ^Aa^	2.46 ± 0.11 ^Ac^	3.28 ± 0.23 ^Bb^
Tibia			
1 day old	1.19 ± 0.12 ^Ca^	0.88 ± 0.04 ^Cb^	1.05 ± 0.09 ^Dab^
10 days old	2.47 ± 0.23 ^Ba^	1.56 ± 0.06 ^Bb^	1.57 ± 0.08 ^Cb^
21 days old	3.95 ± 0.37 ^Aa^	2.08 ± 0.17 ^Ac^	3.12 ± 0.11 ^Ab^
24 days old	4.06 ± 0.24 ^Aa^	2.04 ± 0.09 ^Ac^	2.68 ± 0.17 ^Bb^
Rib			
1 day old	0.26 ± 0.04 ^Cab^	0.17 ± 0.01 ^Bb^	0.29 ± 0.05 ^Ba^
10 days old	0.42 ± 0.02 ^Ba^	0.22 ± 0.02 ^Bb^	0.27 ± 0.01 ^Bb^
21 days old	0.57 ± 0.05 ^Aa^	0.29 ± 0.02 ^Ab^	0.51 ± 0.02 ^Aa^
24 days old	0.56 ± 0.04 ^Aa^	0.30 ± 0.02 ^Ac^	0.45 ± 0.03 ^Ab^
Lumbar vertebrae			
1 day old	0.31 ± 0.03 ^Cab^	0.24 ± 0.03 ^Cb^	0.33 ± 0.03 ^Ca^
10 days old	0.55 ± 0.05 ^Ba^	0.38 ± 0.02 ^Bb^	0.33 ± 0.04 ^Cb^
21 days old	0.83 ± 0.08 ^Aa^	0.48 ± 0.05 ^Ab^	0.75 ± 0.04 ^Aa^
24 days old	0.91 ± 0.06 ^Aa^	0.45 ± 0.02 ^ABc^	0.64 ± 0.05 ^Bb^

Values are presented as means with SE (*n* = 10). Mean values with different lowercase superscript letters in the same row indicate significant differences (*p* < 0.05) among different pig breeds at the same age, while mean values with different uppercase superscript letters in the same column indicate significant differences (*p* < 0.05) among different ages of the same breed. XB, Xiangcun black; TB, Taoyuan black.

**Table 6 animals-15-03536-t006:** Changes in bone mineral density of piglets before and after weaning (g/cm^3^).

Item	Duroc Piglet	XB Piglet	TB Piglet
Femur			
1 day old	0.17 ± 0.01 ^C^	0.17 ± 0.01 ^B^	0.18 ± 0.01 ^B^
10 days old	0.21 ± 0.01 ^B^	0.21 ± 0.02 ^A^	0.19 ± 0.01 ^B^
21 days old	0.25 ± 0.01 ^Aa^	0.21 ± 0.01 ^Ab^	0.25 ± 0.01 ^Aa^
24 days old	0.26 ± 0.01 ^Aa^	0.20 ± 0.01 ^ABb^	0.22 ± 0.01 ^Ab^
Tibia			
1 day old	0.16 ± 0.01 ^C^	0.16 ± 0.01 ^B^	0.17 ± 0.01 ^B^
10 days old	0.22 ± 0.01 ^B^	0.21 ± 0.01 ^A^	0.18 ± 0.01 ^B^
21 days old	0.24 ± 0.01 ^Aa^	0.21 ± 0.01 ^Ab^	0.24 ± 0.01 ^Aa^
24 days old	0.24 ± 0.01 ^Aa^	0.20 ± 0.01 ^Ab^	0.22 ± 0.01 ^Ab^
Rib			
1 day old	0.06 ± 0.00 ^Bb^	0.10 ± 0.00 ^a^	0.09 ± 0.01 ^Ca^
10 days old	0.11 ± 0.00 ^A^	0.10 ± 0.01	0.10 ± 0.01 ^BC^
21 days old	0.10 ± 0.00 ^Ab^	0.11 ± 0.01 ^b^	0.12 ± 0.00 ^Aa^
24 days old	0.11 ± 0.01 ^A^	0.11 ± 0.00	0.11 ± 0.01 ^AB^
Lumbar vertebrae			
1 day old	0.11± 0.01 ^B^	0.12 ± 0.01	0.12 ± 0.01 ^BC^
10 days old	0.11 ± 0.01 ^B^	0.13± 0.01	0.10 ± 0.01 ^C^
21 days old	0.12 ± 0.01 ^ABb^	0.12 ± 0.01 ^b^	0.15 ± 0.01 ^Aa^
24 days old	0.14 ± 0.01 ^Aa^	0.12 ± 0.00 ^b^	0.13 ± 001 ^ABa^

Values are presented as means with SE (*n* = 10). Mean values with different lowercase superscript letters in the same row indicate significant differences (*p* < 0.05) among different pig breeds at the same age, while mean values with different uppercase superscript letters in the same column indicate significant differences (*p* < 0.05) among different ages of the same breed. XB, Xiangcun black; TB, Taoyuan black.

**Table 7 animals-15-03536-t007:** Changes in plasma Ca and P levels of piglets before and after weaning (mmol/L).

Item	Duroc Piglet	XB Piglet	TB Piglet
Calcium			
1 day old	2.92 ± 0.07 ^Aa^	2.65 ± 0.06 ^b^	2.45 ± 0.08 ^Bb^
10 days old	2.56 ± 0.09 ^B^	2.61 ± 0.08	2.64 ± 0.07 ^AB^
21 days old	2.39 ± 0.06 ^Bb^	2.55 ± 0.04 ^b^	2.80 ± 0.12 ^Aa^
24 days old	2.61 ± 0.10 ^B^	2.68 ± 0.07	2.62 ± 0.06 ^AB^
Phosphorus			
1 day old	2.44 ± 0.14 ^Ba^	2.07 ± 0.11 ^Cb^	1.78 ± 0.11 ^Cb^
10 days old	3.10 ± 0.08 ^A^	3.16 ± 0.15 ^A^	3.15 ± 0.11 ^A^
21 days old	2.92 ± 0.14 ^Ab^	2.69 ± 0.13 ^Bb^	3.41 ± 0.15 ^Aa^
24 days old	2.46 ± 0.19 ^B^	2.26 ± 0.09 ^C^	2.16 ± 0.08 ^B^

Values are presented as means with SE (*n* = 10). Mean values with different lowercase superscript letters in the same row indicate significant differences (*p* < 0.05) among different pig breeds at the same age, while mean values with different uppercase superscript letters in the same column indicate significant differences (*p* < 0.05) among different ages of the same breed. XB, Xiangcun black; TB, Taoyuan black.

**Table 8 animals-15-03536-t008:** Changes in plasma ALP level of piglets before and after weaning (U/L).

Item	Duroc Piglet	XB Piglet	TB Piglet
1 day old	2518.60 ± 543.63 ^A^	1820.60 ± 388.86 ^A^	1850.60 ± 184.11 ^A^
10 days old	1279.20 ± 155.18 ^Bab^	1403.20 ± 171.08 ^Aa^	926.80 ± 113.03 ^Bb^
21 days old	572.80 ± 73.26 ^BC^	573.60 ± 100.37 ^B^	657.20 ± 62.13 ^BC^
24 days old	361.20 ± 51.80 ^C^	366.60 ± 27.44 ^B^	420.00 ± 37.79 ^C^

Values are presented as means with SE (*n* = 10). Mean values with different lowercase superscript letters in the same row indicate significant differences (*p* < 0.05) among different pig breeds at the same age, while mean values with different uppercase superscript letters in the same column indicate significant differences (*p* < 0.05) among different ages of the same breed. ALP, alkaline phosphatase; XB, Xiangcun black; TB, Taoyuan black.

## Data Availability

The dataset generated and/or analyzed during the current study is available from the corresponding authors upon reasonable request.

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
