# Peer review of "Bone Growth and Development in Different Breeds of Piglets at the Early Age Is Associated with Mineral Deposition"

_animals, 2025, doi:10.3390/ani15243536_

Round 1

Reviewer 1 Report

Comments and Suggestions for Authors

Dear Authors,

This study (animals-3971833) compares early bone development among pig breeds, including Duroc and Chinese native breeds (Taoyuan Black, Xiangcun Black), focusing on bone growth rate, mineral density, strength, and calcium–phosphorus content. It also examines the impact of weaning stress and explores nutritional strategies to enhance skeletal development, particularly in native breeds with slower bone growth. In order to improve the manuscript, the following issues need to be improved:

2.1. Animal and Experimental Design

Line 90- There is no description of the randomization procedure or the criteria used for animal selection (e.g., whether the piglets originated from different litters and, if so, how many, or whether they were randomly selected from the farm).
-The environmental conditions under which the animals were kept, including the rearing system, ambient temperature, and feed composition, are not described.
-The sex of the piglets is not specified, even though sex hormones are known to have a significant influence on bone growth and development.
-Furthermore, information regarding the diet of the sows and the detailed composition of the starter feed is missing, both of which could have influenced the mineral development of the offspring.

-What was the birth weight of the animals studied? The birth weight of the animals studied is an important factor influencing subsequent growth and bone development. The lack of such information makes it difficult to interpret the results related to growth and development, especially when comparing different breeds.

 2.3. Determination of Bone Parameters

Line 114 - There is no detailed description of the methodology used for testing bone strength. The authors refer to the publication by Li et al. (2024), which reports experiments conducted in pigs but also lacks a precise description of the method used to determine bone breaking load. Moreover, Li et al. themselves refer to the study by Jämsä et al. (1998), in which the method was originally developed for mice, further complicating the assessment of its suitability for use in larger animals. It remains unclear what support span was applied during the measurements, especially given that the tested bones differ in length—was the same span used for all samples? The results include only a single parameter (bone strength), while other essential mechanical indicators, such as Young’s modulus, elastic limit, or work to fracture, are not reported. Furthermore, no justification is provided for the selection of the mechanical testing parameters, particularly the preload (5.6 N) and preload velocity (21 mm/min), whereas most studies typically employ a preload velocity of 10 mm/min.

Therefore, the entire methodological description requires clarification to allow proper assessment of the reliability and comparability of the obtained results.

2.6. Statistical Analysis

Line 130 -The authors used a one-way ANOVA, although the data have a two-factor design (breed × age). A two-way ANOVA would be required to reliably assess the effects of age and breed. Tests for normality and homogeneity of variance were not reported.

Line 183 3.2- Changes in Bone Length, Bone Mineral Content, and Bone Density of Piglets - It rather refers to Bone Strength.

It would have been interesting to conduct correlation studies with the calcium/phosphorus levels in bones on BMD or BMC. Such analyses might be able to clarify whether differences in bone mineral content (especially Ca and P) relate to a difference in bone strength and mineralization parameters. Determination of such relationships would enable us to obtain more knowledge about the physiological regulation of bone development and mineral metabolism in relation to breed or age.

  1. Discussion.

The discussion is relatively limited in literature references; therefore, it would be worthwhile to expand it by incorporating an analysis of similar studies, which would help place the results within a broader scientific context. It would also be appropriate to consider the genetic factors of the three breeds examined, as they may significantly influence nutrient absorption, metabolism, and growth potential. Genetic background determines intestinal morphology and functionality, the expression of nutrient transporters, and the efficiency of calcium and phosphorus utilization. Commercial breeds generally exhibit higher feed efficiency and greater mineral absorption capacity compared to native Chinese breeds, which, in turn, show slower growth but likely greater resilience to environmental stress. Considering these determinants could help explain the observed breed-related differences in bone mineralization and growth rate.

Best regards,

Author Response

2.1. Animal and Experimental Design

Line 90− There is no description of the randomization procedure or the criteria used for animal selection (e.g., whether the piglets originated from different litters and, if so, how many, or whether they were randomly selected from the farm).

Response: A total of 30 litters of newborn TB, XB, and Duroc piglets (10 litters/breed) from their respective sows (sows with 9-11 piglets/litter) were randomly selected for this study. At 1, 10, 21 (weaning), and 24 (three days of post-weaning) day-old, a total of 30 piglets (half male and half female) and 10 piglets from each breed (one piglet from each litter), close to the average body weight of the litter, were randomly selected for sampling at each age stage (Lines 90−110).

−The environmental conditions under which the animals were kept, including the rearing system, ambient temperature, and feed composition, are not described.

Response: Experimental piglets were reared in the same barn with different pig breeds separately. The ambient environment for lactating piglets was maintained at a thermostatically controlled temperature of 30 °C for the first week, then reduced by 2 °C the following week until weaning. All piglets had free access to maternal milk until weaning (21 day-old) and starter feed after weaning at 8:00, 12:00, 14:00, and 17:00 daily. The composition of the starter feed is shown in Supplementary Table 1. The experimental house was maintained at a control temperature (23−26 °C) and with forced-air ventilation. The did piglets not receive any vaccinations during the trial (Lines 97−105).

−The sex of the piglets is not specified, even though sex hormones are known to have a significant influence on bone growth and development.

Response: We agree with the reviewer comment. To minimize the sex biasness, we conducted this experiment with half males and half females. We have this statement in the revised manuscript (Lines 107−110).

−Furthermore, information regarding the diet of the sows and the detailed composition of the starter feed is missing, both of which could have influenced the mineral development of the offspring.

Response: To compare differences in bone growth and development among three pig breeds, all experimental piglets were fed a similar diet formulated to meet the Chinese local swine nutrient requirements (GB NY/T 65 2005), and the premixes met the National Research Council (NRC, 2012) diet requirements. Furthermore, we added a detailed composition of the starter feed in Supplementary Table S1 (Lines 100−104).

−What was the birth weight of the animals studied? The birth weight of the animals studied is an important factor influencing subsequent growth and bone development. The lack of such information makes it difficult to interpret the results related to growth and development, especially when comparing different breeds.

Response: Thanks for your suggestion. We have added the average birth weight of piglets of different piglets has been included in the revised manuscript (Lines 91−94).

2.3. Determination of Bone Parameters

Line 114− There is no detailed description of the methodology used for testing bone strength. The authors refer to the publication by Li et al. (2024), which reports experiments conducted in pigs but also lacks a precise description of the method used to determine bone breaking load. Moreover, Li et al. themselves refer to the study by Jämsä et al. (1998), in which the method was originally developed for mice, further complicating the assessment of its suitability for use in larger animals. It remains unclear what support span was applied during the measurements, especially given that the tested bones differ in length—was the same span used for all samples? The results include only a single parameter (bone strength), while other essential mechanical indicators, such as Young’s modulus, elastic limit, or work to fracture, are not reported. Furthermore, no justification is provided for the selection of the mechanical testing parameters, particularly the preload (5.6 N) and preload velocity (21 mm/min), whereas most studies typically employ a preload velocity of 10 mm/min.

Therefore, the entire methodological description requires clarification to allow proper assessment of the reliability and comparability of the obtained results.

Response: Thanks for your detailed comments and suggestions. We apologize for an inappropriate citation of the methodology. We have further cross-checked the methodology for the bone breaking load evaluation and corrected accordingly. To determine the bone breaking load, a quasi-static three-point bending test was performed using a universal testing machine (Zwick Roell, Ulm, Germany) under the following parameters: a support span equal to 40% of the average bone length (rounded to the nearest millimeter) and a loading rate of 10 mm/min until fracture. The peak of maximum force was recorded (Lines 125−128).

2.6. Statistical Analysis

Line 130−The authors used a one-way ANOVA, although the data have a two-factor design (breed × age). A two-way ANOVA would be required to reliably assess the effects of age and breed. Tests for normality and homogeneity of variance were not reported.

Response: Thanks for your valuable comments and suggestions.

Lines 144−153: The results are presented as means with SE values. Data normality test was assessed using the Shapiro-Wilk test, and homogeneity of variance was evaluated using Levene’s test. The individual piglets were considered the experimental unit for all analyses. Statistical analyses were performed by a two-way ANOVA for pig breed and day of age using R software (version 4.2.1, R Core Team, 2022). Differences between the means of the experimental groups were assessed using the one-way ANOVA and Tukey’s post-hoc test for comparison. Differences among groups were considered significant when P-values were < 0.05. Correlations between Ca and P levels in bones on BMC and BMD were analyzed using Spearman’s correlation test with the R program.

Line 183 3.2− Changes in Bone Length, Bone Mineral Content, and Bone Density of Piglets - It rather refers to Bone Strength.

Response: Thanks for your correction. We have corrected it in the revised manuscript (Line 204).

It would have been interesting to conduct correlation studies with the calcium/phosphorus levels in bones on BMD or BMC. Such analyses might be able to clarify whether differences in bone mineral content (especially Ca and P) relate to a difference in bone strength and mineralization parameters. Determination of such relationships would enable us to obtain more knowledge about the physiological regulation of bone development and mineral metabolism in relation to breed or age.

Response: We appreciate your valuable suggestions. We have further performed correlation analyses to clarify whether differences in bone mineral content are associated with differences in bone strength and mineralization parameters. We added the relevant findings as Figure 3 and a detailed description in the results and discussion.

  1. Discussion.

The discussion is relatively limited in literature references; therefore, it would be worthwhile to expand it by incorporating an analysis of similar studies, which would help place the results within a broader scientific context. It would also be appropriate to consider the genetic factors of the three breeds examined, as they may significantly influence nutrient absorption, metabolism, and growth potential. Genetic background determines intestinal morphology and functionality, the expression of nutrient transporters, and the efficiency of calcium and phosphorus utilization. Commercial breeds generally exhibit higher feed efficiency and greater mineral absorption capacity compared to native Chinese breeds, which, in turn, show slower growth but likely greater resilience to environmental stress. Considering these determinants could help explain the observed breed-related differences in bone mineralization and growth rate.

Response: Thank you for your insightful suggestions. Due to the limited previous literature on bone growth and development of different breeds of pigs, it is challenging to have a fruitful discussion by comparing with similar studies. We have further improved the discussion section, incorporating the possible underlying mechanism with available resources associated with the current findings. Indeed, genetic background attributes, including body weight at birth, feed conversion efficiency, intestinal morphology, barrier function, digestibility, nutrient transport, and mineral absorption efficiency, are the key factors in bone growth and development. We have discussed these points in the revised manuscript for better clarity.

Reviewer 2 Report

Comments and Suggestions for Authors

The manuscript # animals-3971833, entitled “Bone Growth and Development in Different Breeds of Piglets at the Early Age is Associated with Mineral Deposition” by Md. Abul Kalam Azad et al. investigated differences in bone growth and development among different pig breeds during early growth stages. Due to the following concerns, I think the current manuscript needs a major  revision.

  1. Their results showed that Chinese native piglets have slower bone growth than Duroc piglets, but Taoyuan black piglets exhibited higher mineral deposition compared to Xiangcun black piglets. It is generally believed that cross-bred pigs have hybrid advantages. In this study, the cross-bred XB piglets showed a decrease in bone growth and development compared to the Duroc and TB pigs. What are the potential reasons for this? It is recommended to add them to the discussion.
  2. This manuscript emphasized the challenge of weaning for pigs and explored the differences in bone growth and development between different breeds of pigs before and after weaning. However, the author does not discuss what effects of weaning on the bone growth and development between in different breeds of pigs or underlying mechanism.
  3. Diet has a significant impact on the bone growth and development in pigs, especially in the early stages. The manuscript only mentions that feeding and management practices were followed by the company's breeding management model. It is recommended to supplement the diet composition and nutritional components, especially calcium and phosphorus levels, which can help explain the effect of dietary mineral content on the bone growth and development of piglets.
  4. Table 1, the position of “Lumbar vertebrae” was different from others index. Please double check.

Author Response

The manuscript # animals-3971833, entitled “Bone Growth and Development in Different Breeds of Piglets at the Early Age is Associated with Mineral Deposition” by Md. Abul Kalam Azad et al. investigated differences in bone growth and development among different pig breeds during early growth stages. Due to the following concerns, I think the current manuscript needs a major revision.

1. Their results showed that Chinese native piglets have slower bone growth than Duroc piglets, but Taoyuan black piglets exhibited higher mineral deposition compared to Xiangcun black piglets. It is generally believed that cross-bred pigs have hybrid advantages. In this study, the cross-bred XB piglets showed a decrease in bone growth and development compared to the Duroc and TB pigs. What are the potential reasons for this? It is recommended to add them to the discussion.

Response: Thank you for your valuable comments and suggestions. We have added more detailed discussion to improve the clarity in the revised manuscript.

Lines 369−376: Previous research evidence has indicated that weaning stress due to changes in diet, such as from mother milk to solid feed, induces inflammatory processes in the small intestine and hinders nutrient and mineral absorption, leading to deficiencies in mineral metabolism [1]. Moreover, our previous findings also indicated that TB pigs exhibit greater immune tolerance than XB pigs at weaning, which may be due to the Chinese native breed having better adaptation to the local environment compared to the cross-breed [2]. The above-mentioned features may be the possible reasons for decreased bone growth in XB piglets at weaning.

[1] Heaney, R.P.; Layman, D.K. Amount and type of protein influences bone health. Am J Clin Nutr. 2008, 87, 1567S−1570S. https://doi.org/10.1039/ajcn/87.5.1567S.

[2] Ding, S.; Cheng, Y.; Azad, M.A.K.; Zhu, Q.; Huang, P.; Kong, X. Development of small intestinal barrier function and underlying mechanism in Chinese indigenous and Duroc piglets during suckling and weaning periods. Anim Nutr. 2024, 16, 429−442. https://doi.org/10.1016/j.aninu.2023.09.005.

2. This manuscript emphasized the challenge of weaning for pigs and explored the differences in bone growth and development between different breeds of pigs before and after weaning. However, the author does not discuss what effects of weaning on the bone growth and development between in different breeds of pigs or underlying mechanism.

Response: Thanks for your valuable comments. We have further revised the discussion with more emphasis on the weaning effects on bone growth and development between different breeds of pigs, and also added the underlying mechanism (Lines 379−385, 400−407, 427−431).

3. Diet has a significant impact on the bone growth and development in pigs, especially in the early stages. The manuscript only mentions that feeding and management practices were followed by the company's breeding management model. It is recommended to supplement the diet composition and nutritional components, especially calcium and phosphorus levels, which can help explain the effect of dietary mineral content on the bone growth and development of piglets.

Response: Thanks for your valuable comments and suggestions. To compare differences in bone growth and development among three pig breeds, all experimental piglets were fed a similar diet. The experimental diet was formulated to meet the Chinese local swine nutrient requirements (GB NY/T 65 2005), and the premixes met the National Research Council (NRC, 2012) diet requirements. Furthermore, we added a detailed composition of the starter feed in Supplementary Table S1.

  1. Table 1, the position of “Lumbar vertebrae” was different from others index. Please double check.

Response: Thanks for your correction. We corrected it in the revised manuscript.

Reviewer 3 Report

Comments and Suggestions for Authors

Comments to the Authors

  1. Lines 32−34: Provide more details about the experimental piglets. The piglets were obtained from how many sows? What were the litter sizes?
  2. Lines 65−68: Whether the newborn piglets or the early age of piglets? Provide better clarification for this statement.
  3. Lines 90−94: This section is too concise. Provide more details about the origin of the experimental animals. Piglets from how many sows? What were the litter sizes of those sows? How were the piglets selected? All piglets from the same litter or different litters?
  4. Lines 90−94: Provide more details, did all pig breeds have the same feeding regime, or correspond to their diet recommendation? What about the vaccination programs for the experimental piglets? How about the rearing environment conditions?
  5. Lines 97−99: The amount of blood sample?
  6. Lines 100−101: Provide a brief description of the anesthesia and exsanguination process.
  7. Line 105: Not dried, it should be removed of moisture. Correct this statement.
  8. Line 125: The levels of…
  9. Lines 129−132: What was the experimental unit for statistical analysis?
  10. Lines 142−144: Change “During 21−24 day-old” to “At 21 and 24 day-old” for better clarity. Cross-check throughout the text.
  11. Line 184: As listed in Table 4.
  12. Lines 187−188: …the bone strength of femur…; correct these types of errors throughout the text.
  13. Lines 221−223: …the BMD of ribs…
  14. Lines 250−256: Clearly specify the sub-figures (A and B) in the text for better clarification.
  15. Lines 289−293: What are the changes for diets and stressors? A brief detail is necessary for better clarification.
  16. Line 330: BW
  17. Lines 356−373: Regarding plasma-bone Ca and P levels, a possible biological mechanism would strengthen the discussion.
Comments on the Quality of English Language

Some grammatical revisions would enhance the text's clarity.

Author Response

1. Lines 32−34: Provide more details about the experimental piglets. The piglets were obtained from how many sows? What were the litter sizes?

Response: We added more details in the revised manuscript (Lines 32−33).

2. Lines 65−68: Whether the newborn piglets or the early age of piglets? Provide better clarification for this statement.

Response: Thanks for your correction. We have corrected it in the revised manuscript (Lines 66−67).

3. Lines 90−94: This section is too concise. Provide more details about the origin of the experimental animals. Piglets from how many sows? What were the litter sizes of those sows? How were the piglets selected? All piglets from the same litter or different litters?

Response: We have added more details in the revised manuscript.

Lines 90−94: A total of 30 litters of newborn TB, XB, and Duroc piglets (10 litters/breed) with an average birth weight of the piglet (TB, 1.50 ± 0.25 kg; XB, 1.39 ± 0.19 kg; and Duroc, 1.82 ± 0.43 kg) from their respective sows (sows with 9-11 piglets/litter) were randomly selected for this study. At 1, 10, 21 (weaning), and 24 (three days of post-weaning) day-old, a total of 30 piglets (half male and half female) and 10 piglets from each breed (one piglet from each litter), close to the average body weight (BW) of the litter, were randomly selected for sampling at each age stage.

4. Lines 90−94: Provide more details, did all pig breeds have the same feeding regime, or correspond to their diet recommendation? What about the vaccination programs for the experimental piglets? How about the rearing environment conditions?

Response: Thanks for your valuable comment. To compare differences in bone growth and development among three pig breeds, all experimental piglets were fed a similar diet. The experimental diet was formulated to meet the Chinese local swine nutrient requirements (GB NY/T 65 2005), and the premixes met the National Research Council (NRC, 2012) diet requirements. Furthermore, we added a detailed composition of the starter feed in Supplementary Table S1.

The ambient environment for lactating piglets was maintained at a thermostatically controlled temperature of 30 °C for the first week, then reduced by 2 °C the following week until weaning. After weaning, the experimental house was maintained at a control temperature (23−26 °C) and with forced-air ventilation.

Furthermore, the experimental piglets did not receive any vaccinations during the trial.

5. Lines 97−99: The amount of blood sample?

Response: Blood samples (10 mL) were collected from the anterior vena cava of each piglet and transferred into heparin anticoagulant tubes for further processing (Lines 110−111).

6. Lines 100−101: Provide a brief description of the anesthesia and exsanguination process.

Response: We have added more details of the anesthesia and exsanguination process in the revised manuscript (Lines 113−114).

7. Line 105: Not dried, it should be removed of moisture. Correct this statement.

Response: Thanks. We have corrected it.

8. Line 125: The levels of…

Response: Corrected.

9. Lines 129−132: What was the experimental unit for statistical analysis?

Response: The individual piglets are considered as the experimental unit for statistical analysis (Line 147).

10. Lines 142−144: Change “During 21−24 day-old” to “At 21 and 24 day-old” for better clarity. Cross-check throughout the text.

Response: Thanks for your correction. We have crosscheck throughout text and corrected accordingly.

11. Line 184: As listed in Table 4.

Response: Thanks. Corrected.

12. Lines 187−188: …the bone strength of femur…; correct these types of errors throughout the text.

Response: Thanks for your corrections. We corrected this type of error throughout the text.

13. Lines 221−223: …the BMD of ribs…

Response: Corrected.

14. Lines 250−256: Clearly specify the sub-figures (A and B) in the text for better clarification.

Response: We have cited the sub-figures (A and B) in the revised text.

15. Lines 289−293: What are the changes for diets and stressors? A brief detail is necessary for better clarification.

Response: We have added brief details of these stressors in the revised manuscript (Lines 338−341).

16. Line 330: BW

Response: Corrected.

17. Lines 356−373: Regarding plasma-bone Ca and P levels, a possible biological mechanism would strengthen the discussion.

Response: Thanks for your valuable suggestion. We have further added the possible biological mechanism regarding the plasma-bone Ca and P levels to clarify the discussion.

Round 2

Reviewer 1 Report

Comments and Suggestions for Authors

Dear Authors,

The description of the methodology in a scientific paper should be thorough, as it allows other researchers to replicate the study and verify the results. A precise description is crucial for the credibility and scientific nature of the work, allowing for the assessment of the appropriateness of the chosen methods for the research objectives, and enabling the identification of any methodological limitations. The authors have made modifications to the manuscript, but there are still several issues that require further clarification.

“The midpoint diameter of the femur, tibia, and rib was measured using a Vernier caliper. Afterward, a quasi-static three-point bending test was performed using a universal testing machine (Zwick Roell, Ulm, Germany) under the following parameters: a support span equal to 40% of the average bone length (rounded to the nearest millimeter) and a loading rate of 10 mm/min until fracture. The peak of maximum force was recorded.

What was the purpose of measuring the midpoint diameter of the femur, tibia, and rib? Was the data used in the 3-point bending test? What specific Zwick Roell machine model was used to test the bones? What was the measuring range of the testing machine's head (e.g., 1 N to 10 kN). What software was used to analyze the testing data?

It seems to me that these methodological inconsistencies should be addressed and clearly explained in the version of the manuscript submitted for further processing.

Best regards, 

Author Response

Comments:

The description of the methodology in a scientific paper should be thorough, as it allows other researchers to replicate the study and verify the results. A precise description is crucial for the credibility and scientific nature of the work, allowing for the assessment of the appropriateness of the chosen methods for the research objectives, and enabling the identification of any methodological limitations. The authors have made modifications to the manuscript, but there are still several issues that require further clarification.

“The midpoint diameter of the femur, tibia, and rib was measured using a Vernier caliper. Afterward, a quasi-static three-point bending test was performed using a universal testing machine (Zwick Roell, Ulm, Germany) under the following parameters: a support span equal to 40% of the average bone length (rounded to the nearest millimeter) and a loading rate of 10 mm/min until fracture. The peak of maximum force was recorded.

What was the purpose of measuring the midpoint diameter of the femur, tibia, and rib? Was the data used in the 3-point bending test? What specific Zwick Roell machine model was used to test the bones? What was the measuring range of the testing machine's head (e.g., 1 N to 10 kN). What software was used to analyze the testing data?

It seems to me that these methodological inconsistencies should be addressed and clearly explained in the version of the manuscript submitted for further processing.

Response: We appreciate the reviewer’s time and insightful comments on our revised manuscript again. Indeed, a precise description of the methodology is crucial for the credibility and scientific nature of the work.

To obtain an accurate compression curve for the designated software, the diameters at the midpoint of the femur, tibia, and ribs were determined. Afterward, these data were used for the calculation. According to the instrument protocols, the midpoint diameter data were required to be entered before the test was performed.

The Zwich Roell Z5.0 universal testing machine with a measuring range of 1 N to 5000 N was used for this study. NexygenePlus 4.18 materials testing software was used to analyze the test data.

We have added all of these details in the Methods section of the revised manuscript (Lines 124−130).

Reviewer 2 Report

Comments and Suggestions for Authors

No more questions

Author Response

Comment: No more questions

Response: We would like to thank the reviewer again for your positive feedback.